# Flare7K: A Phenomenological Nighttime Flare Removal Dataset

**Yuekun Dai    Chongyi Li    Shangchen Zhou    Ruicheng Feng    Chen Change Loy**
S-Lab, Nanyang Technological University
{YDAI005, chongyi.li, s200094, ruicheng002, ccloy}@ntu.edu.sg
https://Nukaliad.github.io/projects/Flare7K

## Abstract

Artificial lights commonly leave strong lens flare artifacts on images captured at night. Nighttime flare not only affects the visual quality but also degrades the performance of vision algorithms. Existing flare removal methods mainly focus on removing daytime flares and fail in nighttime. Nighttime flare removal is challenging because of the unique luminance and spectrum of artificial lights and the diverse patterns and image degradation of the flares captured at night. The scarcity of nighttime flare removal datasets limits the research on this crucial task. In this paper, we introduce, Flare7K, the first nighttime flare removal dataset, which is generated based on the observation and statistics of real-world nighttime lens flares. It offers 5,000 scattering and 2,000 reflective flare images, consisting of 25 types of scattering flares and 10 types of reflective flares. The 7,000 flare patterns can be randomly added to flare-free images, forming the flare-corrupted and flare-free image pairs. With the paired data, we can train deep models to restore flare-corrupted images taken in the real world effectively. Apart from abundant flare patterns, we also provide rich annotations, including the labeling of light source, glare with shimmer, reflective flare, and streak, which are commonly absent from existing datasets. Hence, our dataset can facilitate new work in nighttime flare removal and more fine-grained analysis of flare patterns. Extensive experiments show that our dataset adds diversity to existing flare datasets and pushes the frontier of nighttime flare removal.

## 1  Introduction

Lens flare is an optical phenomenon in which intense light is scattered or reflected in an optical system. It leaves a radial-shaped bright area and light spots on the captured photo. The effects of flares are more severe in the nighttime environment due to multiple artificial lights. This phenomenon may lead to low contrast and suppressed details around the light sources, degrading the image's visual quality and the performance of vision algorithms. Taking nighttime driving with stereo cameras as an example, the scattering flare may be misestimated as close obstacles by stereo matching algorithms. For aerial object tracking, the bright spots introduced by the lens flare may mislead the algorithm to track flares rather than flying objects [15]. To avoid these potential risks raised by lens flare, the mainstream approach is to optimize the hardware design, such as using specially-designed lens group or applying anti-reflective coating. Although professional lenses can mitigate the flare effect, they cannot solve the inherent problem of flare. Besides, fingerprints, dust, and wear in front of the lens often bring unexpected flare that cannot be alleviated by hardware, especially in smartphone and monitor imaging. A flare removal algorithm is thus highly desired.

Typical flares can be broadly categorized into three major types: *scattering flare*, *reflective flare*, and *lens orb (a.k.a. backscatter)* [27, 11, 9]. We distinguish these three flares according to their response to the light source movement. *Scattering flares* are caused by dust and scratches on the lens. This type of flare will produce radial line patterns. When moving the lens or the light source, the scattering will always wrap around the light source and keep the same pattern on the captured photo. *Reflective flares* are caused by multiple reflections between air-glass interfaces in a lens system [9]. Their patterns are determined by the shape of the aperture and lens structure. Such patterns often manifest as a series of circles and polygons on the captured photo [10]. Different from scattering flares, when moving the camera, reflective flares will move in the direction opposite to the light source. *Lens orbs*

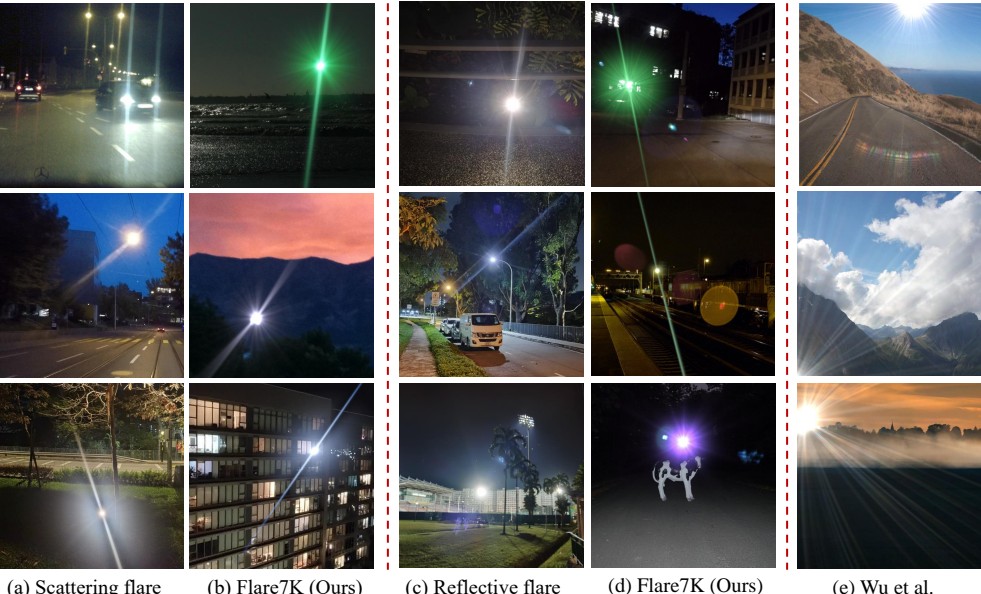

| (a) Scattering flare | (b) Flare7K (Ours) | (c) Reflective flare | (d) Flare7K (Ours) | (e) Wu et al. |

Figure 1: Nighttime photography often suffers from different types of lens flares. In (a), streaks and radial stripes are caused by scattering flare. In (c), bright blobs and large rings in deep blue are caused by reflective flare. (b) and (d) depict some of our synthetic nighttime flare-corrupted images. (e) show some flare-corrupted images synthesized by Wu et al. [27]. Images of reflective flares are taken with the rear camera of the Huawei P40 smartphone. Parts of images of scattering flares are obtained from Dark Zurich [20] and NightOwls [14] nighttime driving dataset. In contrast to Wu et al. [27], our flare data is more similar to real-world nighttime data.

are induced by unfocused particles of dust or drops on the lens surface [6]. They are aperture-shaped polygons fixed in the same position in photography. Only the lens orbs around the light source will be lightened, and they will not move with the light source or camera motion. Because the position of lens orbs is relatively fixed, this effect is much easier to be removed in a video [11]. Thus, we mainly focus on the removal of scattering flares and reflective flares at nighttime in this study.

Removing nighttime flares is extremely challenging. First, the flares' patterns are diverse, caused by the varied location and spectrum of the light source, unstable defects in the lens manufacture, and random scratches and greasy dirt during everyday utilization. Second, the dispersion of light at different wavelengths and interference between small optical structures can also lead to rainbow-like halo and colored moiré. Although there are some traditional flare removal methods [2, 1, 24], they mainly concentrate on detecting and wiping off small bright blobs in the reflective flare. Recently, some learning-based flare removal methods [27, 16, 22, 5] have been proposed for daytime flare removal or removing the flare with a specific type of pattern. The scarcity of paired nighttime flare-corrupted/flare-free data limits the development of deep learning-based nighttime flare removal.

Large-scale data is indispensable for training deep models. Some methods try to collect paired flare-corrupted/flare-free images. In Wu et al.'s method [27], physically-based flares and flare photos were taken in the darkroom and overlaid on flare-free images, forming paired data. Sun et al. [22] assumed that all flares are generated with the same 2-point star Point Spread Function (PSF). As a result, all these generated flares in Sun et al.'s method are relatively homogeneous. Unlike daytime flares, the street lights' luminance is significantly lower than the sunshine. Hence, the defects in the lens manufacture will only lead to tiny scattering flares that are always acceptable in nighttime scenes. Thus, professional cameras can always capture a clear night view in this case. However, for monitor lenses, smartphone cameras, UAVs, and autonomous driving cameras, the fingerprint, daily wear, and dust may function as a grating, thus resulting in the streaks (strip-shaped flares) that are still obvious at night. Besides, the spectrum of artificial lights is quite different from the sunshine and may introduce different diffraction patterns. The gap in the flare pattern between daytime and nighttime makes the models trained on daytime flare datasets scarcely perform well at night.

To facilitate the research on nighttime flare removal, we built a large-scale layered flare dataset with elaborately designed night flares, called Flare7K, the first dataset of its kind. It offers 5,000 scattering and 2,000 reflective flare images, consisting of 25 types of scattering flares and 10 types

of reflective flares. All flare patterns in our dataset are synthesized based on the observation and statistics of real-world night flares. Since scattering and reflective flare are independent, we generate the respective flare data separately. Thus, different reflective flares can be added to any scattering flare to achieve richer diversity. The 7,000 flare patterns can be randomly added to flare-free images, forming the flare-corrupted and flare-free image pairs that can be used for training deep models. In Figure 1, we present the comparison between our data and the real-world nighttime images. We also present the synthetic data from a recent flare dataset [27]. In comparison, our data is more similar to the real-world nighttime images in terms of the flare patterns. Besides, each scattering flare image in our dataset can be divided into three parts: light source, streak, and glare. The separation of flare components makes our dataset more interpretable and manipulatable than previous flare datasets [27].

The main contributions of this study are three-fold. 1) We construct the first large-scale Flare7K dataset, providing a valuable benchmark to facilitate the research on this challenging nighttime flare removal task. 2) Our dataset offers rich annotations, catalyzing new research not only in nighttime flare removal, but also in flare component segmentation, light source extraction, and reflective flare detection. 3) Extensive experiments demonstrate that our nighttime flare dataset can help solve the nighttime flare removal problem better than existing methods and datasets.

## 2 Related work

**Lens flare dataset.** Collecting a large-scale paired flare-corrupted and flare-free image dataset needs tedious human labor. To solve this issue, Wu et al. [27] proposed a semi-synthetic flare dataset, which is the only flare dataset in the literature. It comes with 2,001 captured flare images and 3,000 simulated flare images. These flare images will be added to flare-free images to simulate flare-corrupted situations. However, all the captured flare photos are taken by the same camera and under the same light source within the same distance. The homogeneous setting makes the captured flare images look pretty similar and have a limited effect on removing the flares with diverse kinds of lenses and light sources. Besides, Wu et al.'s physically-realistic flares also have a considerable gap with real-world nighttime flares, as the comparison shown in Figure 1. Lens flare simulation algorithms [9, 10] have been studied for a long time and are already widely used in visual effects (VFX) of films, animation, and television programs. With the help of Optical Flares (a plug-in for rendering lens flares in Adobe After Effects), we build a realistic lens flares dataset based on real-world reference images, aiming to solve the problems of domain gap and the lack of diversity.

**Single image flare removal.** Prior works mainly focus on veiling glare removal [17, 23] of HDR image in the backlit scene and reflective flare removal that involves saturated blobs [2, 1, 24]. Due to the lack of paired data that contains scattering flares and diverse reflective flares, deep learning-based methods are restricted. Wu et al. [27] proposed a semi-synthetic flare dataset to synthesize flare-corrupted images. With the flare-corrupted image and flare-free image pairs, a pix2pix model based on U-Net [18] was trained to restore the flare-free image. Qiao et al. [16] collected natural flare-corrupted and flare-free images to obtain unpaired flare data. Following the idea of Cycle-GAN [34], Qiao et al. trained a framework with a light source detection module, a flare generation module, a flare detection module, and a flare removal module. Due to the uniqueness of nighttime flares and the scarcity of paired nighttime flare data, existing methods still cannot cope with nighttime flares well.

**Nighttime defogging and nighttime visibility enhancement.** Multiple scattering of light in fog brings the glare effect around the light source. Although the physical principle of this glare effect is different from lens flare, they have similar appearances. These properties make nighttime defogging and flare removal share some common ground. In the early work of glare effect removal [13], it is a mainstream method to calculate PSF for each light source and then use deconvolution to recover the light source image. With the wide application of dark channel prior [7], many nighttime haze removal methods based on statistical priors achieve good performance. Li et al. [12] proposed a new nighttime haze model and separated the glare effect by using gray world assumption and smooth glare prior. To achieve computational efficiency, Zhang et al. [31] proposed a fast nighttime haze removal structure by using maximum reflectance prior. Yan et al. [28] noticed that grayscale images are less affected by multiple colors of atmospheric light, thus using the grayscale component to guide a neural network to remove haze. Zhang et al. [32] designed a new nighttime haze synthetic method that can mimic light rays and object reflectance and then used the synthetic data to train a haze removal network. Besides nighttime defogging, recent nighttime visibility enhancement methods also begin to focus on light effect suppression. In Sharma et al.'s high dynamic range nighttime image enhancement module [21], a noise and light effect suppression network was introduced to extract low-frequency light effect based on the gray world assumption for the glare-free component. All these methods assume that the glare effect is smooth, and hence cannot remove the high-frequency component in scattering flares. Therefore, suppressing glare effect properly in nighttime images is still an open problem.

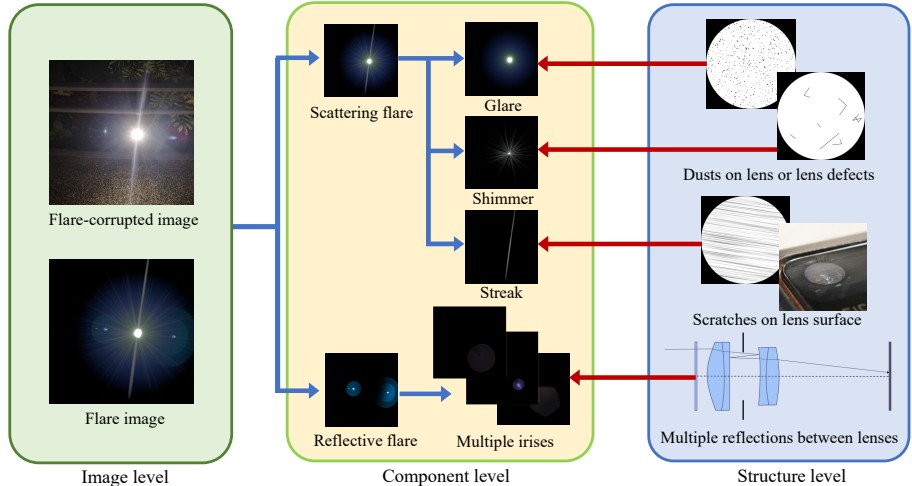

Figure 2: Formulation of nighttime lens flare. Lens flare can be viewed as a combination of scattering flare and reflective flare. Multiple reflections between lens surfaces lead to a line of irises pattern, and reflective flare can be viewed as the addition of these irises. Scattering flare can be divided into glare, shimmer, and streak. Streak is caused by grating-like scratches in the front of the lenses. Glare and shimmer are brought by the combined action of lens defect and dust on the pupil. Different types of dust make glare and shimmer have different patterns.

# 3 Physics on nighttime lens flare

Typical nighttime lens flares, i.e., scattering flare and reflective flare, are complex as they comprise many components including halos, streaks, irises, ghosts, bright lines, saturated blobs, haze, glare, shimmer, sparkles, glint, spike balls, rings, hoops, and caustic. In VFX, computational photography, optics, and photography, an identical type of components may have different names. To avoid confusion, we group these names into several common types based on their patterns. For instance, sparkles, glint, and spike balls are all radial line-shaped patterns. In this paper, we use shimmer to represent all these types of radial line-shaped components. To facilitate a better understanding of our proposed dataset, we explain the formation principle of each type of nighttime lens flare as follows.

## 3.1 Scattering flare

The common components in scattering flares can be divided into glare, shimmer, and streak. Their patterns are shown in Figure 2. Glare is a smooth haze-like effect around the light source, also known as the glow effect [8]. Even in an ideal lens system, the pupil with a limited radius will still function as a low pass filter, resulting in a blurry light source. Moreover, abrasion or dotted impurities in the lens will lead to the lens' uneven thickness, noticeably increasing the area of the glare effect. Besides, dispersion makes the hue of the glare not globally constant. As shown in Figure 2, the pixels of the glare far away from the light source are bluer than the pixels around the light source. During daytime with sufficient illumination, the scene around the light is bright enough to cover the glare effect. However, in a low-light condition, the glare is significantly brighter than the scene, hence cannot be ignored for nighttime flare removal.

Shimmer (a.k.a., sparkles, glint, spike balls) is a pattern with multiple radial stripes caused by the aperture's shape and line-shaped impurities and lens defects [9]. Due to the structure of the aperture, the pupil is not a perfect round, thus producing a star-shaped flare. Taking the dodecagon-shape aperture as an example, diffraction around the edge of the aperture projects a point light source to a dodecagram on the photo. Different from the aperture, line-shaped lens defects always lead to uneven shimmer. For the lens flare in the daytime, as a light source with high intensity, the sun will leave bright shimmers over the whole screen. In contrast, the intensity of the artificial light is lower and the area of the shimmer is always similar to the glare effect. Since shimmer is only different from glare in terms of pattern, it can also be viewed as a high-frequency component of the glare.

Streaks (a.k.a., bright lines, stripes) are line-like flares that are significantly longer and brighter than shimmer [19, 22]. They often appear in smartphone photography and nighttime driving video. Oriented oil stains or abrasion on the front lens may act as grating and cause beam-like PSF. During the daytime, streaks are just like brighter shimmer. However, in a low-light condition, even a light source with low intensity may generate streaks across the whole screen. Since one cannot always keep a smartphone's lens or vehicle-mounted camera clean, this phenomenon is conspicuous at nighttime.

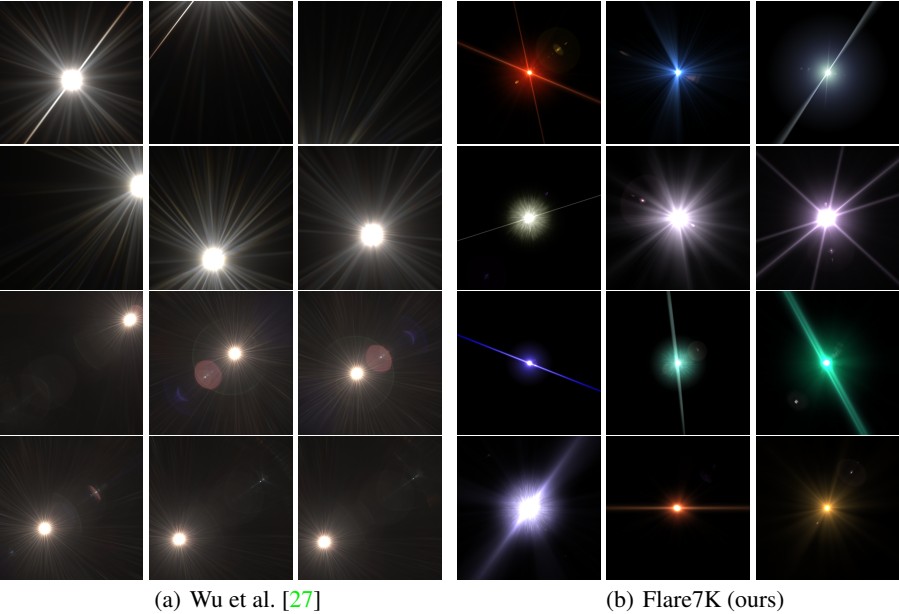

(a) Wu et al. [27]           (b) Flare7K (ours)

Figure 3: Flare pattern comparison between our Flare7K dataset and Wu et al.'s dataset [27].

## 3.2 Reflective flare

Reflective flares (a.k.a., ghosting) are caused by reflection in multiple air-glass lens surfaces [9]. For a lens system with $n$ optical surfaces, even if the light is exactly reflected twice, there are still $n(n-1)/2$ kinds of combinations to choose two surfaces from $n$ surfaces [27, 9]. Generally speaking, the reflective flares form a combination of different patterns like circles, polygons, or rings on the image. Due to multiple reflections between lenses, it is challenging to synthesize reflective flares in physics. To simulate reflective flares, a more straightforward method is to use 2D approaches [10]. Specifically, for 2D reflective flare rendering, since the hoop and ring effect caused by dispersion is not apparent at night, we can abstract the reflective flare as a line of different irises as shown in Figure 2. During the process of reflection, if the light path is blocked by the field diaphragm, this would result in a clipping iris. In 2D approaches, this effect can be simulated by setting a clipping threshold for the distance between the optical center and the light source. If this distance is longer than the clipping threshold, parts of the irises would be clipped proportionally.

Ideally, each iris can be added to the image independently. Moreover, there will not be interference between different irises. However, in real-world scenes, the neighboring rays are often correlated and generate triangle mesh. To avoid blocking artifacts, Ernst et al. [4] proposed a way for caustics rendering and introduced a technique for combining and interpolating these irises. In our method, since rendering physically realistic caustics increases the difficulty of simulating reflective flare, we use specific caustics patterns to simulate this effect.

## 4 Flare7K dataset

In this section, we first show the differences between our Flare7K dataset and the existing flare dataset [27]. Then, we provide the details about the construction of our dataset.

### 4.1 Comparison with existing flare dataset

The only existing flare dataset is the one proposed by Wu et al. [27], which is mainly designed for daytime flare removal. Thus, the streak effect and glare effect that commonly exist in nighttime flares are not considered in Wu et al.'s dataset. In terms of nighttime flares, the patterns are mainly decided by the stains in front of the lens. The variety of contamination types makes it difficult for physics-based methods such as Wu et al.'s approach to collect real nighttime flares by traversing all different pupil functions. This results in the lack of diversity and the domain gap between synthetic flares in Wu et al.'s dataset and the real-world scenes.

To solve the issues of domain gap and diversity, we propose a new nighttime flare dataset by referring to real nighttime flare images. Specifically, we take hundreds of nighttime flare images with different

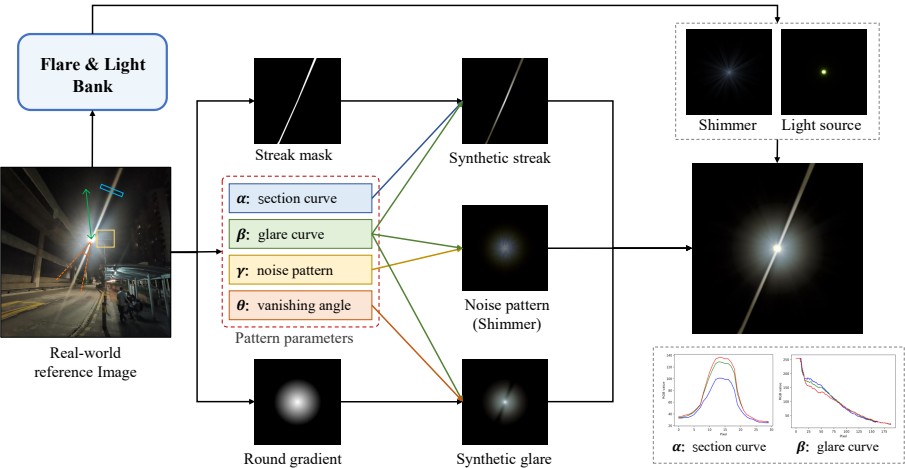

Figure 4: The pipeline of scattering flare synthesis. To synthesize scattering flare, we first obtain streak section curve $\alpha$, glare descent curve $\beta$, noise patch near the light source $\gamma$, and the vanishing corner's angle $\theta$ around the streak from the reference image. $\alpha$ and $\theta$ are used to synthesize the glare effect while $\alpha$ and $\beta$ are used to simulate the streak. To simulate the degradation around the light source, we add a blurred fractal noise pattern on the shimmer to create a realistic flare.

types of lenses (smartphone and camera) and various light sources as reference images. We aggregate the captured images and summarize the scattering flares as 25 typical types based on their patterns. Since reflective flares are directly related to the type of the lens group, we also capture some video clips using different cameras as references for reflective flare synthesis. By referring to these real-world nighttime flare videos, we design a group of irises for each specific kind of camera and synthesize 10 typical types of reflective flares. For each type of flare, we generate 200 images with different parameters such as the glare's radius, the streak's width, etc. We finally obtain 5,000 scattering flares and 2,000 reflective flares. Since flare rendering is relatively mature, we choose to directly use the plug-in Video Copilot's Optical Flares in Adobe After Effects to generate customized flares. As shown in Figure 3, our flare patterns are more diverse and closer to real-world nighttime cases. Furthermore, we compare the differences between our dataset and Wu et al.'s dataset in Table 1. The comparison shows that our new dataset offers richer patterns and annotations, which benefit broader applications, such as lens flare segmentation and light source extraction. More details about these extended applications are presented in the supplementary material.

Table 1: A comparison between our Flare7K dataset and Wu et al.'s dataset. The 'type' indicates the number of different patterns of scattering flare and reflective flare. In particular, we show the numbers as type of scattering flares + type of reflective flares. Since it is difficult to separate shimmer and glare by definition, we provide glare annotations which also contain shimmer effect.

| Dataset | Statistics | | | | Annotations | | | |
|---|---|---|---|---|---|---|---|---|
| | pattern | synthetic | real | type | light source | reflective flare | streak | glare |
| Wu et al. [27] | 5,001 | 3,000 | 2,001 | 2+1 | ✗ | ✗ | ✗ | ✗ |
| Flare7K | 7,000 | 7,000 | 0 | 25+10 | ✓ | ✓ | ✓ | ✓ |

## 4.2 Scattering flare generation

Figure 4 presents our scattering flare synthesis pipeline. We separate the lens flare into four components including shimmer, streak, glare, and light source. For each component, we analyze the parameters like glare's radius range and color-distance curve in reference images. Then, we use Adobe After Effect to synthesize flare templates.

**Glare synthesis.** From the reference flare-corrupted images of each type, we first plot the relationship between the RGB value of the pixel and its distance to the light source as the glare curve. Divided by the glare's radius, such a color-distance relationship can be viewed as a color correction curve. Applying this curve to a round gradient pattern with glare's radius can produce the glare effect of this type of flares. Since the region's luminance around the streak sometimes becomes weaker than the normal area, we measure the vanishing angle manually and use a feathered mask around the streak to

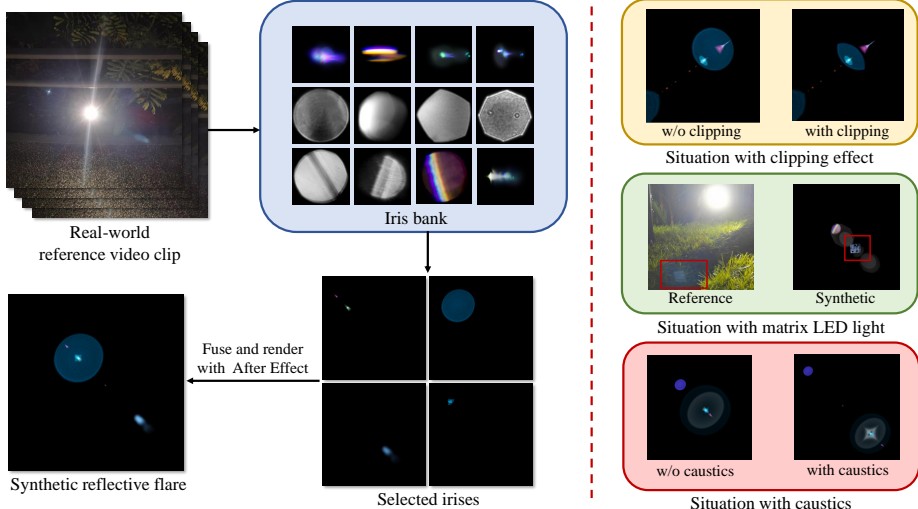

Figure 5: The pipeline of reflective flare synthesis. Since clipping effects and caustics are not obvious in a single image, we capture video clips as references. While synthesizing reflective flares, we first filter most similar irises from Optical Flares Plug-in's iris bank. Then, we manually adjust the position, size, and color of these irises to fit the reference. Finally, these irises are fused to create a reflective flare template in Adobe After Effect. For some special cases like caustics, matrix light, or clipping effect, the details are presented in Section 4.3.

decrease the opacity of glare in these areas. The angle of this missing corner is set to a variable to cover more cases while generating this type of scattering flare.

**Streak synthesis.** In Sun et al. [22]'s work, it assumes that the streak is always generated with a 2-point star PSF. However, the streak effect is not even symmetric, and one side is often much sharper than the other side. To imitate this effect, we manually draw a mask for each type of streak in Adobe After Effect and set the width as a variable. Then, we plot the RGB value of the streak's section and glare section and use this curve to colorize the streak and blur the mask's edge. The blur size for each edge is derived from the section curve's half-life value.

**Shimmer synthesis.** As for shimmer, we use the shimmer template of Optical Flares and adjust the parameters until it roughly matches the flare of the image. In the area around the light source, the image often suffers from strong degradation that is challenging to be simulated by Optical Flares. If we suppose our lens flare is smooth, this degradation would be separated as part of the deflared image. Thus, we use Adobe After Effect's default plug-in fractal noise to generate a noise patch and then add the radial blur effect to it, thus creating a radial noise pattern. This pattern will be added to a shimmer template of Optical Flares to compose a realistic shimmer.

**Light source synthesis.** We apply thresholding on flare-corrupted images to obtain a group of overexposed tiny shapes. To simulate the light source's glow effect, we apply another plug-in named Real Glow on different tiny shapes in Adobe After Effects. To ensure that only the light source region is overexposed, the light source is made larger than the glare's overexposure part. Then, it is added to the flare with screen blend mode. This mode ensures that the overexposed region is not expanded and brings realistic visual results.

### 4.3 Reflective flare generation

For reflective flares, the plug-in Optical Flares' Pro Flares Bundle contains 51 kinds of different captured high-quality iris images that can serve as the iris bank. While comparing with the reference video, we pick the most similar irises and manually adjust their size and color with the Optical Flares plug-in. Since the distances from different irises to a light source are always proportional, we follow the plug-in's pipeline to set different iris components in a line with proportional distance to a light source. After that, we can obtain a reflective flare template.

For some special types of reflective flares, we also consider flares' dynamic triggering mechanisms like caustics and clipping effect. These phenomena will happen when the light source's position on the image is far from the lens' optical center. As stated in Section 3.2, the caustics phenomenon is

caused by interference between different irises. To simulate this effect, we use Optical Flares' default caustics template to generate a caustics pattern in the center of the iris. To simulate the dynamic triggering effect, the opacity of this caustics pattern is set to be proportional to the distance between the iris and the light source. As for the clipping effect, it is generated when the reflected light path is blocked by more than two lenses' apertures. It can be viewed as the intersection of two irises. Thus, when the iris-light distance is larger than the clipping threshold, we start to erase parts of the iris by using another iris as a mask. This iris will only serve as a mask and will not be rendered.

In nighttime situations, matrix LED light is common and may bring lattice-shaped reflective flare. To imitate this effect, we synthesize some irises in the shape of the lattice as shown in Figure 5. Compared to the previous dataset [27], these designs reflect real-world nighttime situation better.

### 4.4  Paired test data collection

Since there is existing no nighttime flare removal dataset, we collect our own synthetic and real nighttime flare paired data for evaluation with full-reference image quality assessment metrics.

**Synthetic test data collection.** We synthesize nighttime flare data using our proposed pipeline as introduced in Section 4, in which both flare images and flare-free images do not appear in our training dataset. To make the synthetic test data more realistic, we first take the same scene with the rear camera of Huawei P40 (smartphone camera) and Sony $\alpha$ 6400 with Sigma 16mm F1.4 (professional camera). To imitate real-world scenarios, we do not deliberately clean the smartphone's lens. In this way, the images taken by the smartphone are always flare-corrupted while the images taken by the professional camera are not. Referring to these flare-corrupted images, we synthesize flare images and add them to the images captured by the professional camera to compose flare-free/flare-corrupted pairs. At last, we synthesize 100 pairs of data for the test.

**Real test data collection.** For most well-designed lenses, the flares of a nighttime scene are caused by the stains on the lens's surface (or scratches on the windshield for nighttime driving). To reproduce these flares, we use fingers and a cloth to wipe the front lens of the camera to mimic common stains. After that, we use lens tissue to clean the front lens slightly to obtain flare-free images. The action of cleaning may still cause a tiny misalignment of the paired images. Thus, we align the paired images manually and obtain 100 pairs of real-world flare-corrupted/flare-free images as our real-world test dataset. Since the ground truth may still be influenced by the slight flares brought by the lens's defects, the evaluations on paired real-world data can only be used as a reference. It cannot fully reflect the actual performance of flare removal methods.

## 5  Experiments

To demonstrate the effectiveness and advantages of our dataset, we compare the performance of different datasets and methods for nighttime flare removal. We also present a benchmark of the existing image restoration methods on our dataset.

**Experimental setting.**  In the absence of nighttime flare removal methods, we compare our baseline model with a nighttime dehazing method [32], a nighttime visual enhancement method [21], and a flare removal method [27]. Zhang et al. [32] propose a nighttime haze synthetic method that can simulate light rays. With the synthetic data, they trained a network. We use its pre-trained model for comparison. Sharma et al.'s nighttime visual enhancement method [21] is an unsupervised test-time training method that can extract low-frequency light effects based on gray world assumption and glare-smooth prior. We follow its setting to test the results on our test data. Wu et al.'s study [27] is most related to our work. Thus, we take it as a baseline. Since Wu et al. do not provide their model checkpoint, we use their released code and data to train a model for comparison. With the proposed dataset, we also follow Wu et al. [27] and train a U-Net [18] as a baseline network.

To facilitate the related research on our Flare7K dataset, we build a benchmark for more state-of-the-art learning-based image restoration methods, including HINet [3] , MPRNet [30], Restormer [29], and Uformer [25]. Following Wu et al. [27]'s pipeline, we train these models with the image size of $512 \times 512$ with a batch size of 2. All these experiments are conducted on Nvidia Geforce RTX 3090. Due to the limited GPU memory (24G memory) of Nvidia Geforce RTX 3090, we reduce the parameters of the MPRNet and Restormer that adopt heavy network structures. More experimental details, settings, and qualitative comparisons are presented in the supplemental material. The quantitative results are presented in Table 2.

**Qualitative comparison.** We first show the visual comparison of real-world nighttime flares in Figure 6. The comparison suggests that recent approaches for nighttime haze removal [32] and

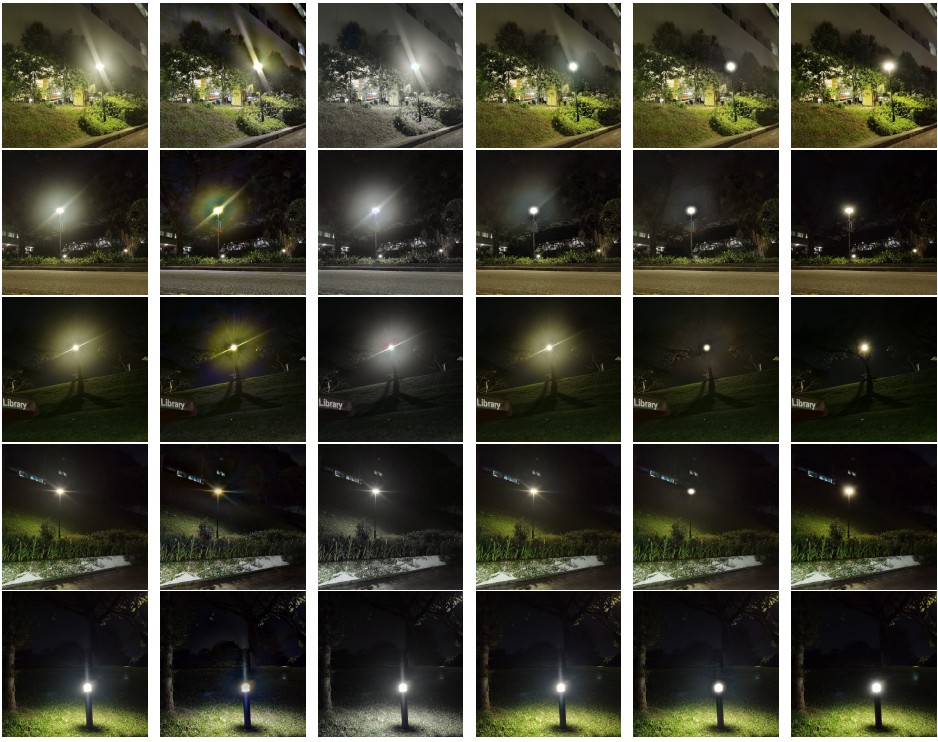

(a) Real input  (b) Zhang [32] (c) Sharma [21]  (d) Wu [27]       (e) Ours         (f) GT

Figure 6: Visual comparison of flare removal on real-world nighttime flare images.

nighttime visual enhancement [21] have little effect on nighttime lens flare. In contrast, the model trained on our dataset can produce satisfactory outputs on nighttime flare-corrupted images. Since the only difference between Wu et al. [27] and our method is the training dataset, the visual results show that our dataset can better characterize nighttime flares. We show more results of our method in Figure 7. In these challenging cases, our approach yields satisfactory flare-free results. The results of different methods on synthetic nighttime flare images and different image restoration networks' performances are shown in the supplemental material.

Table 2: Quantitative comparison of synthetic and real nighttime flare-corrupted data. Since Wu et al. [27] also use U-Net [18] as the backbone network, it shows that our dataset has better performance on real nighttime flare-corrupted images. The benchmark of the image restoration methods for nighttime flare removal is listed on the right part of the table. "*" denotes models with reduced parameters due to the limited GPU memory. It is expected that their original models would perform better. More details are presented in the supplementary material.

| Data\Method | | Input | Previous work | | | Network trained on our dataset | | | | |
|---|---|---|---|---|---|---|---|---|---|---|
| | | | Zhang [32] | Sharma [21] | Wu [27] | U-Net [18] | HINet [3] | MPRNet* [30] | Restormer* [29] | Uformer [25] |
| Real-world | PSNR↑ | 22.56 | 21.02 | 20.49 | 24.61 | 26.11 | 26.74 | 26.14 | 26.28 | **26.98** |
| | SSIM↑ | 0.857 | 0.784 | 0.826 | 0.871 | 0.879 | 0.882 | 0.878 | 0.883 | **0.890** |
| | LPIPS↓ | 0.078 | 0.174 | 0.112 | 0.060 | 0.055 | 0.048 | 0.050 | 0.054 | **0.047** |
| Synthetic | PSNR↑ | 22.77 | 21.04 | 20.01 | 27.88 | 29.07 | 29.97 | 29.87 | 29.45 | **30.47** |
| | SSIM↑ | 0.921 | 0.841 | 0.865 | 0.952 | 0.958 | 0.959 | 0.959 | 0.950 | **0.965** |
| | LPIPS↓ | 0.060 | 0.136 | 0.111 | 0.031 | 0.022 | 0.021 | 0.020 | 0.025 | **0.017** |

**Quantitative comparison.** We use full-reference metrics PSNR, SSIM [26], and LPIPS [33] to quantify the performance of different methods in Table 2.   Since Sharma et al.'s method [21] is based on the gray world assumption, it fails in the glare effect in white. Zhang et al.'s method [32] is mainly designed for nighttime haze. Although it can alleviate the lens flare, it also changes the image's color, leading to the decrease in SSIM. In Table 2, the only difference between U-Net [18] on our dataset and Wu et al.'s method is training data, suggesting the effectiveness of Flare7K. In comparison, Uformer [25] performs best on our task. All these networks achieve good performance, revealing the reliability of our dataset.

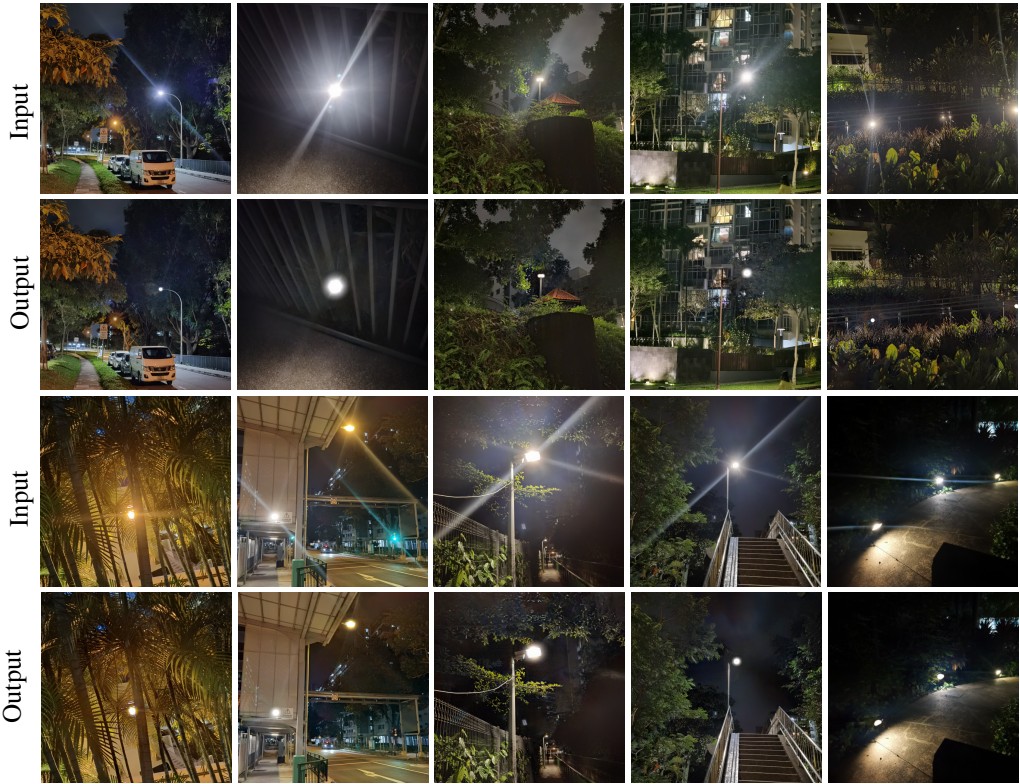

Figure 7: Our results on real-world nighttime flare-corrupted images. Since our dataset contains diverse flares, it can generalize well to different situations.

# 6   Limitations

The experiments above show that Flare7K can better represent flares in the nighttime situation. However, the networks trained on our data may still fail to address some challenging cases. Since reflective flares always produce bright spots similar to bright windows and street lights in the distance at night, these windows or lights may also be removed as a part of reflective flares. Besides, when the light source is close to the camera, it may leave a large glare that may cover the whole image. This kind of glare is tough to be removed. In this situation, the streak and the region around the light source may get saturated in one or more channels. Existing image decomposition-based methods cannot complete this missing information well. These limitations are mainly caused by the flare removal method rather than our dataset. To solve these problems, one would need to consider semantic priors as input or introduce better network structures.

# 7   Conclusion

We have presented a new dataset, aiming at advancing nighttime flare removal. Our dataset contains 5,000 scattering and 2,000 reflective flare images, consisting of 25 types of scattering flares and 10 types of reflective flares. Our dataset focuses on common nighttime flare components like streak and glare. Besides, we provide rich annotations for different flare components. With this dataset, one can train nighttime flare removal networks to improve the quality of nighttime images and boost the stability of nighttime vision algorithms. Extensive experiments show that our dataset is sufficiently diverse to facilitate the removal of different types of nighttime lens flares.

**Acknowledgement:** This study is supported under the RIE2020 Industry Alignment Fund – Industry Collaboration Projects (IAF-ICP) Funding Initiative, as well as cash and in-kind contribution from the industry partner(s). It is also partially supported by the NTU NAP grant.

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
