# OpenReview forum: "Flare7K: A Phenomenological Nighttime Flare Removal Dataset"
_NeurIPS.cc/2022/Track/Datasets_and_Benchmarks — NeurIPS 2022 Datasets and Benchmarks _

### Official Review · Reviewer_9BDD · 2022-07-12
**Significant improvement and thorough analysis**

**Rating:** 8
**Confidence:** 4
**Correctness:** The dataset was constructed in a soun…
**Clarity:** The paper was well written.

**Strengths:**

The presented dataset has not only been shown to have already introduced significant boosts to model performance but has also opened up doors to other areas of research because of the accompaniment of annotation data (although it is not as immediately clear why it would be relevant to industrial usage). The paper has also done a very thorough literature review into the topic of image flare that will aid future studies.

The dataset is also diverse across geographical regions. It isn't highlighted in the paper, but the website for the dataset showcases images from the dataset that are taken from countries outside of Singapore and in different environments.

**Weaknesses:**

The most immediate potential shortcoming I see to their method is that flare may have been synthetically generated in images where there would not otherwise be any flare even with any configuration of camera and lens. For example, there seem to be images in the dataset where flares were synthetically generated even though the original image does not have any lights or there were multiple lights in the image but only flares for one of them were generated. Just based on this disparity, there is a clear distribution shift between their dataset and real-world data which might have held back the models' abilities to generalize.

The paper could include details of methods to collect images that flares were synthetically generated for as well as details of how the Adobe After Effects tools worked under the hood.

**Additional Feedback:**

Well done. Great contribution overall.

**Documentation:**

The landing page for the dataset (where presumably most will go to access the data) provides sufficient details but misses out on licensing information.

**Ethics:**

The authors actively avoided including personal information in their dataset.

**Relation To Prior Work:**

Yes. Prior work was clearly discussed.

**Summary And Contributions:**

The paper presents a dataset of synthetically generated flare images based on real-world flare images collected from different cameras with different lenses. Using the real-world reference as well as a strong background in the physics of flares, the synthetically generated dataset had a much greater diversity than prior works thus allowing models to generalize better on unseen data. The paper has shown that a model trained on their dataset has shown undeniable improvements than when trained on datasets from prior works.

---

> ### Author Response · Authors · 2022-08-26
> **Response to Assigned Reviewer 9BDD**
>
> We appreciate the positive and constructive comments on our paper. The raised concerns are addressed as follows. We also updated our main paper and supplementary material.
>
> > *The most immediate potential shortcoming I see to their method is that flare may have been synthetically generated in images where there would not otherwise be any flare even with any configuration of camera and lens. For example, there seem to be images in the dataset where flares were synthetically generated even though the original image does not have any lights or there were multiple lights in the image but only flares for one of them were generated. Just based on this disparity, there is a clear distribution shift between their dataset and real-world data which might have held back the models' abilities to generalize.*
>
> Thank you so much for the valuable comments. Since light sources may appear in any places in real-world situations (like the moon in the sky, garden lamp on the grass, lighthouse on the sea), it is possible for lens flare to appear in some special scenes. Thus, we consider it is reasonable to synthesize flare-corrupted images with a synthetic flare image with a light source and a randomly chosen natural image.
>
> What's more, since we synthesize the flare-corrupted/flare-free pairs on the fly, it is easy to change this generation pipeline to fit real-world situations. For example, we can add the synthetic lens flares only on the exposed region to make sure that there is a light source. This trick can help us to solve this distribution shift. For multiple light situations, we can also add multiple flare images of different sizes based on each exposed region's area to each light source. This also implies the potential of our dataset.
>
> > *The paper could include details of methods to collect images that flares were synthetically generated for as well as details of how the Adobe After Effects tools worked under the hood.*
>
> The reference flare-corrupted image collection methods are stated in lines 201-202. While pointing the camera to light sources like street light, it will leave a lens flare on the image. To simulate real-world situation, we do not adopt special metrics and skills while capturing these photos.  If the reviewer is interested in our reference flare-corrupted dataset, we are glad to provide a link to download it.
>
> We introduce the pipeline of our flare generation methods in Section 4.2 and Section 4.3. Adobe After Effects is just like Adobe Photoshop for video. Here is a tutorial about the use of the Adobe After Effect's plug-in:
>
> 	https://www.videocopilot.net/products/opticalflares/features/plug-in_overview/.
>
> > *The landing page for the dataset (where presumably most will go to access the data) provides sufficient details but misses out on licensing information.*
>
> Thank you for your reminder. We have added the licensing information CC BY-NC-SA 4.0 of our dataset on the homepage.

---

> > ### Comment · Reviewer_9BDD · 2022-08-28
> > **Response**
> >
> > > For example, we can add the synthetic lens flares only on the exposed region to make sure that there is a light source.
> >
> > I'm not sure what you mean by this. Perhaps could you instead clarify if a single light source would only create one flare? And how do you position the flares in your synthetic images?
> >
> > > the moon in the sky, garden lamp on the grass, lighthouse on the sea
> >
> > These make sense to me. But it wouldn't make sense if you had a picture of a plain wall that shouldn't have any source of light but still has flare generated for it. A real-world image with flare would show some source of light like a candle or lamp.

---

> > > ### Author Response · Authors · 2022-08-29
> > > **Response to Assigned Reviewer 9BDD**
> > >
> > > > > For example, we can add the synthetic lens flares only on the exposed region to make sure that there is a light source.
> > > >
> > > > I'm not sure what you mean by this. Perhaps could you instead clarify if a single light source would only create one flare? And how do you position the flares in your synthetic images?
> > >
> > > Thank you for your reply! This sentence means that our flare generation method allows creating one flare on an existing single light source. It can be achieved by locating the light source region by detection algorithms. However, in our provided default dataloader function, we randomly add light source and flare (the light source and flare are added simultaneously) on the flare-free images. We found that the 'random' manner as a kind of data augmentation can enrich the diversity of training data, thus improving the capability of deep models.
> > >
> > > We agree that adding the flare to the exiting light source region could generate more realistic flare images. However, it may need to collect more diverse light source images. Considering the limited publicly available light source images, our random manner is easier to be achieved by only using existing flare-free images. As suggested, besides the current generation method that randomly adds light source and flare, we also provide another solution that first locates the light source region of a flare-free image with light source and then adds flare only on the light source region. However, we need to collect more images with light source for this method. The corresponding results and datalader function of this solution can be found at https://github.com/ykdai/Flare7K/blob/main/Generate_flare_on_light/generate_flare_loc.ipynb .
> > >
> > >
> > >
> > > > > the moon in the sky, garden lamp on the grass, lighthouse on the sea
> > > >
> > > > These make sense to me. But it wouldn't make sense if you had a picture of a plain wall that shouldn't have any source of light but still has flare generated for it. A real-world image with flare would show some source of light like a candle or lamp.
> > >
> > >  In addition, a picture of a plain wall can have both light source and flare (such as bright LED night light like [this type](https://firebasestorage.googleapis.com/v0/b/firescript-577a2.appspot.com/o/imgs%2Fapp%2FResearch_NK%2FuDTe8OOEwu.png?alt=media&token=8bb6c42f-9fc3-4369-b6fc-908b1eaa6313)) despite it being infrequent. For infrequent cases, our 'random' flare generation manner can solve the long-tail distribution issue.
> > >
> > > At last, we wish to emphasize that the experiments demonstrate the effectiveness of our current random generation manner for processing real nighttime flares.

---

### Official Review · Reviewer_GQCe · 2022-07-25
**Interesting work. The authors introduce the task and dataset in detail.**

**Rating:** 7
**Confidence:** 3
**Correctness:** In L223, datase -> dataset

**Strengths:**

[1] The paper provides a detailed introduction, which makes readers easily understand why lens flare occurs, and why lens flare is a significant problem.

[2] Provide the first nighttime flare dataset, Flare7K, and show the effectiveness of Flare7K.

**Weaknesses:**

[1] Question: Are authors planning to provide the flare-corrupted and flare-free image pairs or provide the training code that generates paired images on the fly?
In my opinion, the same paired dataset used in the experiments is necessary to reproduce the results and fairly compare methods.
I downloaded and checked the datasets.
The dataset only contains 7K flare patterns and test data, not flare-corrupted and flare-free image pairs.

Furthermore, I cannot find the number of paired data. It would be important to grasp how many paired images are required to remove nighttime flare.
In supplementary materials, I find "To train our nighttime flare removal model, our paired flare-corrupted and flare-free images are generated on the fly." and "60 epochs on Flickr24K".
Does it mean the number of paired data is 24K x 60?

\

[2] Hard to understand section 4.2 dataset generation and Figure 4.
From L188-L191, "Figure 4 presents our scattering flare synthesis pipeline. From the reference flare images of each type, we can obtain the relationship between the RGB value of the pixel and its distance to the light source. Such a relationship can be viewed as a color correction curve"

Question 1: Is the reference image in Figure 4 a "real" nighttime flare image?

Question 2: If so, how do we obtain the steak mask, 4 pattern parameters, and round gradients from the reference image? "we can obtain the relationship" seems not enough to explain.

In addition, a brief explanation of Optical Flares in Adobe After Effects is necessary.

\

[3] The evaluation set only contains 20 synthetic and 20 real paired data.
It is small. (but at least it seems enough to compare flare removal methods, as shown in Table 2.)
Increasing the size of the test set (especially real data) would be better.
Rewriting L223 to L239 would be better.
It is hard to know that the test set consists of 20 real and 20 synthetic paired data at a glance.

\

[4] It would be better if there were more details about other methods Zhang [40] and Sharma [39].
From Table 2, their performances are worse than "input". I wonder why and how they work.
I find slightly more details in supplementary materials, but I still cannot roughly understand how they work.
In addition, in my opinion, when using Zhang's method, authors should fine-tune their model on Flair7K rather than using the pre-trained model.
Please ignore this comment if Zhang's official code does not provide a proper training code.

\

[5] According to L107-L108, Qiao et al. [8] tried to remove lens flare with unpaired data using the cycle-GAN framework.
If possible, I would like to see the results when the cycle-GAN framework is trained on unpaired data generated by Flare7K.
Please ignore this comment if additional experiments are prohibited or somewhat burdensome.

**Additional Feedback:**

Nice demo page and video. They help understand this work.

**Clarity:**

In L177, what do "these issues" indicate?

In L227, it is better to notify that Huawei P40 is a smartphone, and the Sony α 6400 with Sigma 16mm F1.4 is a professional camera.

**Documentation:**

Need to include "Datasheet" in the supplementary materials.

**Relation To Prior Work:**

Yes

**Summary And Contributions:**

This work provides the first nighttime flare removal dataset, Flare7K.
Before this work, there was only a daytime flare dataset.
The lack of the nighttime flare dataset hinders research on the task of nighttime flare removal.

The dataset contains 7K flare patterns that can be used to synthetically generate flare-corrupted and flare-free pairs.
The authors compare the patterns of the existing daytime dataset (by Wu [7])  and the proposed nighttime dataset (Flare7K) in Figure 3.
Flare7K seems to have more diverse patterns.

Figure 5 shows the effectiveness of Flare7K on the nighttime flare removal task: only replacing the training dataset with Flare7K improves results.
In my opinion, from the qualitative comparison in Figure 5, the difference between Wu's and the authors' is not small.
However, from the quantitative comparison in Table 2, the difference between Wu's and the authors' seems relatively small.

---

> ### Author Response · Authors · 2022-08-27
> **Response to Reviewer GQCe (Part 1)**
>
> We appreciate the positive and constructive comments on our paper. The raised concerns are addressed as follows. We also updated our main paper and supplementary material.
>
> > *(1) Question: Are authors planning to provide the flare-corrupted and flare-free image pairs or provide the training code that generates paired images on the fly? In my opinion, the same paired dataset used in the experiments is necessary to reproduce the results and fairly compare methods. I downloaded and checked the datasets. The dataset only contains 7K flare patterns and test data, not flare-corrupted and flare-free image pairs.*
> >
> > *Furthermore, I cannot find the number of paired data. It would be important to grasp how many paired images are required to remove nighttime flare. In supplementary materials, I find "To train our nighttime flare removal model, our paired flare-corrupted and flare-free images are generated on the fly." and "60 epochs on Flickr24K". Does it mean the number of paired data is 24K x 60?*
>
> Thank you for your useful advice. In our experiment, our paired images are generated on the fly. During training stage, the flare-free images are sampled from 24k Flickr images [1].  The 24k Flickr images can be found here: https://drive.google.com/file/d/1GNFGWfUbgXfELx5fZtjTjU2qqWnEa-Lr. For a fair comparison, we fix the synthetic and real testing data in our experiments.
>
> We agree that proving paired data is more intuitive. However, we recommend generating paired data on the fly during training state. This is because nighttime flare removal is challenging. More diverse training data generated via on the fly could improve the capability of deep models.
>
> If one wants to see the generated flare-corrupted and flare-free image pairs, we provide our flare-corrupted/flare-free pairs generation script with the dataloader function on our Github paper: https://github.com/ykdai/Flare7K. Using the script, one can see and save the generated image pairs.
>
> References:
>
> - [1] Xuaner Zhang, Ren Ng, and Qifeng Chen. "Single image reflection separation with perceptual losses."  in *IEEE Conference on Computer Vision and Pattern Recognition*, 2018.
> - [2] Yicheng Wu, Qiurui He, Tianfan Xue, Rahul Garg, Jiawen Chen, Ashok Veeraraghavan, and Jonathan T. Barron, “How to train neural networks for flare removal.” in *IEEE International Conference on Computer Vision*, 2021.
>
> > *(2) Hard to understand section 4.2 dataset generation and Figure 4. From L188-L191, "Figure 4 presents our scattering flare synthesis pipeline. From the reference flare images of each type, we can obtain the relationship between the RGB value of the pixel and its distance to the light source. Such a relationship can be viewed as a color correction curve"*
> >
> > *Question 1: Is the reference image in Figure 4 a "real" nighttime flare image?*
> >
> > *Question 2: If so, how do we obtain the steak mask, 4 pattern parameters, and round gradients from the reference image? "we can obtain the relationship" seems not enough to explain.*
> >
> > *In addition, a brief explanation of Optical Flares in Adobe After Effects is necessary.*
>
> Thank you for your suggestions. I have rewritten this section to make it easy to follow. For the details of glare generation, the rewritten part is in lines 221-228. Please find it for details.
>
> **A1:** Yes, the reference image in Figure 4 is a real nighttime flare image. We capture it with the rear camera of the smartphone Huawei P40. To avoid confusion, we have already changed the description in this figure.
>
> **A2:** We have provided more details in Section 4.2 and clarified the confusing parts. We appreciate the reviewer's suggestions to improve the readability of this work. For your convenience, we provide the details as follows:
>
> - ​	**Streak mask**: We draw a mask manually for each type of streak in Adobe After Effect and set the width as a variable.
>
> - ​	**Round gradient**: The RGB value of round gradient's formula can be represented as:
>   $$
>   value=\frac{R-r}{R}*(255,255,255)
>   $$
>   where $r$ is the pixel's distance to the light source. $R$ is the glare's radius. It can be estimated from the glare curve.
>
> - ​	**Section curve**: It is plotted in Python. We select two points on each side of the streak to create a line and plot the pixels' value in this line.
>
> - ​	**Glare curve**: Same as section curve.
>
> - ​	**Noise pattern**: We scale up the region around the light source for reference. Then, we follow this patch to synthesize the shimmer manually in Adobe After Effect.
>
> - ​	**Vanishing angle**: We select one point in the light source and select two points on the angle's edge. We then calculate it in Python.
>
> Also, we add a brief explanation of the Optical Flares plug-in in **Lens flare dataset** in Section 2 of the main paper. You can check the updated content in lines 99-103. If you are interested in this plug-in, please refer to the tutorials for more details:
>
> https://www.videocopilot.net/products/opticalflares/features/plug-in_overview/

---

> > ### Author Response · Authors · 2022-08-27
> > **Response to Reviewer GQCe (Part 2)**
> >
> > > (3) The evaluation set only contains 20 synthetic and 20 real paired data. It is small. (but at least it seems enough to compare flare removal methods, as shown in Table 2.) Increasing the size of the test set (especially real data) would be better. Rewriting L223 to L239 would be better. It is hard to know that the test set consists of 20 real and 20 synthetic paired data at a glance.
> >
> > Thank you for your advice. I agree with the opinion that the recent test dataset is relatively small. Collecting real paired data is quite time-consuming, and it takes us several weeks to collect more. As suggested, we have already increased the size of the test data and updated it in our Flare7K dataset. Now, we have 100 synthetic and 100 real paired data. For your convenience, we upload the test dataset here:
> >
> > https://drive.google.com/file/d/1umUN5Wp-E68uCz7Gzc5RIQ_YhdpNHNuZ/view?usp=sharing
> >
> > > (4) It would be better if there were more details about other methods Zhang [40] and Sharma [39]. From Table 2, their performances are worse than "input". I wonder why and how they work. I find slightly more details in supplementary materials, but I still cannot roughly understand how they work. In addition, in my opinion, when using Zhang's method, authors should fine-tune their model on Flair7K rather than using the pre-trained model. Please ignore this comment if Zhang's official code does not provide a proper training code.
> >
> > We appreciate the reviewer's suggestions to improve the readability of this work. To clarify it, we have added more details about Zhang and Sharma in the **Nighttime defogging and nighttime visibility enhancement** of Section 2. We also provided a more detailed explanation of experimental settings in Section 5 of the updated paper. The added content can be found in lines 296-300. For your convenience, we list the updated content as follows.
> >
> > *Zhang et al. propose a nighttime haze synthetic method that can simulate light rays. With the synthetic data, they trained a network. We use its pre-trained model for comparison. Sharma et al.’s nighttime visual enhancement method is an unsupervised test-time training method that can extract low-frequency light effects based on gray world assumption and glare-smooth prior. We follow its setting to test the results on our test data.*
> >
> > The reasons why these methods are ineffective are updated in **Quantitative comparison** of Section 5. The updated content can be found in lines 323-326. For your convenience, we list the updated content as follows.
> >
> > *Since Sharma et al.’s method is based on the gray world assumption, it fails in the glare effect in white. Zhang et al.’s method is mainly designed for nighttime haze. Although it can alleviate the lens flare, it also changes the image’s color, leading to the decrease in SSIM.*
> >
> > In this section, we compare with Zhang et al. to show that our dataset can synthesize the nighttime lens flare effect better. Since Zhang et al. cannot effectively improve the PSNR and SSIM scores, the results show that the previous nighttime haze synthetic method cannot handle the lens flare effect well. This also implies  the significance of our dataset.
> >
> > We also added more recent deep learning-based image restoration methods including HINet [3], Uformer [4], MPRNet [5], and Restormer [6] for comparison. We have detailed the **Experiment Setting** in Section 5 and added the results in Table 2 of the updated paper. More details and visual results can be found in Section 2 of the supplementary material.
> >
> > References:
> >
> > - [3] Liangyu Chen,  Xin Lu, Jie Zhang, Xiaojie Chu, and Chengpeng Chen, "HINet: Half instance normalization network for image restoration." in *IEEE Conference on Computer Vision and Pattern Recognition Workshops*, 2021.
> > - [4] Zhendong Wang, Xiaodong Cun, Jianmin Bao, Wengang Zhou, Jianzhuang Liu, and Houqiang Li, “Uformer: A general u-shaped transformer for image restoration.” in *IEEE Conference on Computer Vision and Pattern Recognition*, 2022.
> > - [5] Syed Waqas Zamir, Aditya Arora, Salman Khan, Munawar Hayat, Fahad Shahbaz Khan, Ming-Hsuan Yang, and Ling Shao,  “Multi-stage progressive image restoration.” in *IEEE Conference on Computer Vision and Pattern Recognition*, 2021.
> > - [6] Syed Waqas Zamir,  Aditya Arora, Salman Khan, Munawar Hayat, Fahad Shahbaz Khan, and Ming-Hsuan Yang, "Restormer: Efficient transformer for high-resolution image restoration." in *IEEE Conference on Computer Vision and Pattern Recognition*, 2022.

---

> > > ### Author Response · Authors · 2022-08-27
> > > **Response to Reviewer GQCe (Part 3)**
> > >
> > > > (5) According to L107-L108, Qiao et al. [8] tried to remove lens flare with unpaired data using the cycle-GAN framework. If possible, I would like to see the results when the cycle-GAN framework is trained on unpaired data generated by Flare7K. Please ignore this comment if additional experiments are prohibited or somewhat burdensome.
> > >
> > > Thank you for the good advice!  Although Qiao et al. proposed a flare removal framework, they did not release the official code. Thus, we cannot compare our network with their method. Once the official code is publicly available, we will update the comparison results.
> > >
> > > > Need to include "Datasheet" in the supplementary materials.
> > >
> > > We have added a datasheet for our dataset. Please check it in Section 7 of the updated supplementary material.

---

> > ### Comment · Reviewer_GQCe · 2022-08-29
> > **Thank you for answering my questions.**
> >
> > All of my questions are addressed in the comments.
> >
> > Especially, I appreciate that the authors provided the code to generate the paired dataset (it is my most concerning) and extended the test set 20 synthetic and 20 real data to 100 synthetic and 100 real data.
> >
> > The results of more recent deep learning-based image restoration methods (added in the revision) are also very useful.
> >
> > I will take them into account for my decision.
> >
> > Thank you for your kind response.

---

> > > ### Author Response · Authors · 2022-08-29
> > > **Responses to Reviewer GQCe**
> > >
> > > Thank you for recognizing our work!
> > > We appreciate your constructive suggestions and precious time. It helps us improve the quality of our work greatly.

---

### Official Review · Reviewer_zAPP · 2022-07-27
**Only dataset available for nighttime flare removal but manuscript needs major revisions in terms of writing, presentation and experiments.**

**Rating:** 6
**Confidence:** 3
**Correctness:** The Dataset's formation seems to be c…

**Strengths:**

It is the the first nighttime flare removal dataset.
More diversity in annotation with respect to light source, reflective flare and streak.
More number of images compared to previous datasets.


**Weaknesses:**

1)Firstly, the manuscript has several typos and sentences are not well structured. Extensive proof reading required.

2)A lot of citations are missing in sections 1 and 3 when explaining concepts. Even section 3 seems to be completely paraphrased from Wu et al.'s work but citations are missing.

3)As there is only 1 other night flare dataset, comparisons are done with only that paper. It is difficult to find reasonable contributions when compared with only 1 paper.

4)The authors claim that compared to the previous dataset, Flare7K has more annotations. But the difference in size between the two datasets is not huge. Other datasets can easily overcome this size with proper data augmentation methods.

5)The authors have trained their dataset with only the network (U-net) from the paper Wu et al. There are several other state-of-the-art models that the authors could have experimented with in comparison tables to show the reliability of this dataset. In short, experiments are not extensive and clear.

6)In line 177: authors start with a new paragraph with the sentence "To solve these issues".  There is no reference to the said issues mentioned in the previous sections. It is not a good practice to skip referencing sections where there is clear verification of a claim or a fact.

7)In line 213: "compared with previous datasets", again missing references.

8)Table 2 comparison is explained in supplementary material but it is not clear as to how those models' parameters were selected for the evaluation and how Flare7K was deployed on those models. Was Flare7K trained or evaluated on all the models? It looks like only fair comparison is with Wu’s paper as they have used the same training model as Flare7K. But what about Zhang and Sharma? Is the Flare7K evaluated on all the models and that’s the result comparison in Table 2? In short, clear explanation answering these questions are needed for fair comparison.

9)Conclusion is vague. Needs to be re-written summarizing and highlighting specific contributions and results of the paper.

**Additional Feedback:**

The authors have worked hard in building a meaningful dataset and the demo video is interesting, but the manuscript as a whole needs major revisions. It looks like the paper was written in a hurry, therefore, it is missing citations and references and has several typos. Moreover, the paper lacks a proper experimental method ( comparison of results not clear) and the experiments performed is not adequate.
Some example typos:
In line 63: street lights are significant lower than the sunshine---> significantly.
In line 117: To avoid confusing, we universally unique name of each component.--> confusion, universally choose?
In line 118: for better understanding our proposed dataset --> of our dataset
In line 223: Datase--> database

**Clarity:**

Paper needs major revision and extensive proof reading as mentioned in weaknesses. Also, check the comments on Additional Feedback.
The dataset generation section 4 is not easy to follow.

**Documentation:**

URL provided.


**Ethics:**

No information provided.

**Relation To Prior Work:**

Literature review and comparison to prior work is few. Some concepts of the paper look like inspired version of previous papers with no citations (Physics Part). There are additional contributions to those previous works as mentioned in the Strengths.

**Summary And Contributions:**

The authors propose a  first-of-its-kind dataset, Flare7K, for nighttime flare removal. The dataset has 5,000 scattering flare images
and 2,000 reflective flare images. In the dataset, the scattering flares are of 25 types and the reflective flares are of 10 types. In addition, the dataset has more annotations with respect to light source, glare with shimmer, reflective flare, and streak than other flare removal datasets. The dataset is trained on U-net and the results are compared with previous datasets.

---

> ### Author Response · Authors · 2022-08-26
> **Response to Reviewer zAPP (Part1)**
>
> We appreciate the comments on our paper. The raised concerns are addressed as follows. We also updated our main paper and supplementary material.
>
> > 1) *Firstly, the manuscript has several typos and sentences are not well structured. Extensive proof reading required.*
>
> Thanks for your careful reading. We have fixed the typos and rewritten the paper.
>
> > 2) *A lot of citations are missing in sections 1 and 3 when explaining concepts. Even section 3 seems to be completely paraphrased from Wu et al.'s work but citations are missing.*
>
> We have added necessary references in Sections 1 and 3 of the updated paper. Besides, we wish to explain that Section 3 is quite different from Wu et al.'s work.  This part introduces the physics principle of nighttime flare and daytime flare's domain gap. It also offers the principle of streak and glare effect which is absent in Wu et al.'s work. In the part of reflective flare, we introduce the caustic effect and clipping effect. We then follow these phenomena to synthesize our Flare7K dataset.
>
> > 3) *As there is only 1 other night flare dataset, comparisons are done with only that paper. It is difficult to find reasonable contributions when compared with only 1 paper.*
>
> Wu et al.'s dataset is the most related to our Flare7K dataset. To our best knowledge, this is the only flare dataset. The lack of such a dataset motivates us to build this nighttime flare dataset. Although Wu et al.' dataset contains several nighttime flare images, it mainly focuses on daytime flare which is quite different from the nighttime flare.
> In comparison to Wu et al.'s dataset, our Flare7K dataset has unique characteristics:
>
> 1)  We focus more on common nighttime flare components like streak and glare to reduce the domain gap. In comparison, Wu et al.'s dataset ignores these effects.
> 2)  Our dataset is based on different types of nighttime flares. It has more types of flares (25+10) than Wu et al. (2+1), thus generalizing well to different kinds of situations as shown in Figure 7 of the main paper.
> 3)  We provide annotations that are absent in Wu et al.'s dataset. In Section 4 of the supplementary material, we show that these annotations are quite useful for related tasks.
>
> Besides, the models trained on Wu et al.'s dataset cannot handle nighttime flare well while our dataset enables good performance on nighttime flare removal. Therefore, such a new dataset is meaningful and contributes to the development of nighttime flare removal.
>
> > 4) *The authors claim that compared to the previous dataset, Flare7K has more annotations. But the difference in size between the two datasets is not huge. Other datasets can easily overcome this size with proper data augmentation methods.*
>
> We wish to emphasize that our Flare7K dataset contains 25 types of scattering flares and 10 types of reflective flares while the previous dataset e.g., Wu et al.'s dataset only contains 3 types of flares. As stated in Section 2 of the supplementary material, flare-corrupted images will be generated on-the-fly with flare images after data augmentation. Thus, the number of types affects the diversity of the dataset, not the size of the dataset. As for the synthetic data, one can generate the data as many as one wants.
>
> Besides more diverse types of flares, we also provide more annotations including light source, streak, and glare with shimmer which are not provided in the previous dataset.

---

> > ### Author Response · Authors · 2022-08-26
> > **Response to Reviewer zAPP (Part2)**
> >
> > > 5. *The authors have trained their dataset with only the network (U-net) from the paper Wu et al. There are several other state-of-the-art models that the authors could have experimented with in comparison tables to show the reliability of this dataset. In short, experiments are not extensive and clear.*
> >
> > Thank you for your constructive comments.  As suggested, we have added more recent image restoration baselines,  including Restormer [4], MPRNet [3], HINet [1], and Uformer [2] in the updated paper. We have detailed the **Experiment Setting** in Section 5 and added the results in Table 2 of the updated paper. For your convenience, we present the quantitative results as follows. More details and visual results can be found in Section 2 of the supplementary material.
> >
> > We wish to explain that the purpose of using the U-net of Wu et al's work as a baseline model is to demonstrate the advantages of our dataset under the same network structure.
> >
> > | Method        |      |       | Real  |       |      |       | Synthetic |       |
> > | ------------- | ---- | ----- | ----- | ----- | ---- | ----- | --------- | ----- |
> > |               |      | PSNR  | SSIM  | LPIPS |      | PSNR  | SSIM      | LPIPS |
> > | Input         |      | 22.56 | 0.857 | 0.078 |      | 22.77 | 0.921     | 0.060 |
> > | U-Net [5]     |      | 26.11 | 0.879 | 0.055 |      | 29.07 | 0.958     | 0.022 |
> > | HINet [1]     |      | 26.74 | 0.882 | 0.048 |      | 29.97 | 0.959     | 0.021 |
> > | MPRNet [3]    |      | 26.14 | 0.878 | 0.050 |      | 29.87 | 0.959     | 0.020 |
> > | Restormer [4] |      | 26.28 | 0.883 | 0.054 |      | 29.45 | 0.950     | 0.025 |
> > | Uformer [2]   |      | 26.98 | 0.890 | 0.047 |      | 30.47 | 0.965     | 0.017 |
> >
> > Reference:
> >
> > - [1] Liangyu Chen,  Xin Lu, Jie Zhang, Xiaojie Chu, and Chengpeng Chen, "HINet: Half instance normalization network for image restoration." in *IEEE Conference on Computer Vision and Pattern Recognition Workshops*, 2021.
> > - [2] Zhendong Wang, Xiaodong Cun, Jianmin Bao, Wengang Zhou, Jianzhuang Liu, and Houqiang Li, “Uformer: A general u-shaped transformer for image restoration.” in *IEEE Conference on Computer Vision and Pattern Recognition*, 2022.
> > - [3] Syed Waqas Zamir, Aditya Arora, Salman Khan, Munawar Hayat, Fahad Shahbaz Khan, Ming-Hsuan Yang, and Ling Shao,  “Multi-stage progressive image restoration.” in *IEEE Conference on Computer Vision and Pattern Recognition*, 2021.
> > - [4] Syed Waqas Zamir,  Aditya Arora, Salman Khan, Munawar Hayat, Fahad Shahbaz Khan, and Ming-Hsuan Yang, "Restormer: Efficient transformer for high-resolution image restoration." in *IEEE Conference on Computer Vision and Pattern Recognition*, 2022.
> > - [5] Olaf Ronneberger, Philipp Fischer, and Thomas Brox, “U-Net: Convolutional networks for biomedical image segmentation.” in *International Conference on Medical Image Computing and Computer-Assisted Intervention*, 2015.
> >
> > > 6) *In line 177: authors start with a new paragraph with the sentence "To solve these issues". There is no reference to the said issues mentioned in the previous sections. It is not a good practice to skip referencing sections where there is clear verification of a claim or a fact.*
> >
> > Thank you for pointing out this issue. In the updated paper, we have rewritten this part. It can be found in line 200 of the updated  paper. It has been revised as *To solve the issues of domain gap and diversity*. Moreover, the reasons for the domain gap and the lack of diversity are stated in lines 193-199.
> >
> > > 7) *In line 213: "compared with previous datasets", again missing references.*
> >
> > In the updated paper, we have added the references of the compared datasets. It is marked in blue and can be found in line 267 of the updated paper.

---

> > > ### Author Response · Authors · 2022-08-26
> > > **Response to Reviewer zAPP (Part3)**
> > >
> > > > 8) *Table 2 comparison is explained in supplementary material but it is not clear as to how those models' parameters were selected for the evaluation and how Flare7K was deployed on those models. Was Flare7K trained or evaluated on all the models? It looks like only fair comparison is with Wu’s paper as they have used the same training model as Flare7K. But what about Zhang and Sharma? Is the Flare7K evaluated on all the models and that’s the result comparison in Table 2? In short, clear explanation answering these questions are needed for fair comparison.*
> > >
> > > We appreciate the reviewer's suggestions to improve the readability of this work. We have provided a more detailed explanation of experimental settings in  Section 5 of the updated paper.  The added content can be found in lines 296-300. For your convenience, we list the updated content as follows.
> > >
> > > *Zhang et al. propose a nighttime haze synthetic method that can simulate light rays. With the synthetic data, they trained a network. We use its pre-trained model for comparison. Sharma et al.’s nighttime visual enhancement method is an unsupervised test-time training method that can extract low-frequency light effects based on gray world assumption and glare-smooth prior. We follow its setting to test the results on our test data.*
> > >
> > > Zhang et al.'s network is based on MobileNet-v2 backbone [6]. It can achieve better visual performance. However, it will also change the color of the image and may not lead to better PSNR and SSIM. Training a U-Net [5] with their synthetic data may not change this fact.
> > >
> > > Sharma et al.'s method is a test-time training method with U-Net [5]. It is a prior-based method and does not need training data.
> > >
> > > Reference:
> > >
> > > - [6] Andrew G. Howard, Menglong Zhu, Bo Chen, Dmitry Kalenichenko, Weijun Wang, Tobias Weyand, Marco Andreetto, and Hartwig Adam, "MobileNets: Efficient convolutional neural networks for mobile vision applications." arXiv:1704.04861,2017.
> > >
> > > > 9) *Conclusion is vague. Needs to be re-written summarizing and highlighting specific contributions and results of the paper.*
> > >
> > > Thank you for the advice. We have rewritten the conclusion and highlighted the contributions and results of our work.
> > >
> > > > ***Clarity:** Paper needs major revision and extensive proof reading as mentioned in weaknesses. Also, check the comments on Additional Feedback. The dataset generation section 4 is not easy to follow.*
> > >
> > > We have rewritten most parts of the paper, especially the Section 4. The generation of nighttime flare is complex and requires multi-disciplinary knowledge. We attempt to make it easy to follow. In the updated paper, we have provided more detailed illustrations and explanations of the generation process.
> > >
> > > > ***Relation To Prior Work:** Literature review and comparison to prior work are few.*
> > >
> > > In the updated paper, we have added more literature reviews and references as well as more comparisons to previous works. We have marked these citations in blue.

---

> > > > ### Comment · Reviewer_zAPP · 2022-09-01
> > > > **Response of Reviewer**
> > > >
> > > > My concerns are addressed by the reviewers and I appreciate the efforts of the authors. And for that I have increased my rating.
> > > >
> > > > Thank You

---

### Official Review · Reviewer_E5H2 · 2022-07-27
**A good synthetic dataset for nighttime flare removal**

**Rating:** 6
**Confidence:** 4
**Correctness:** Yes.

**Strengths:**

1. The first nighttime flare removal dataset with realistic-looking synthetic flares.
2. Additional annotation of flare components.

**Weaknesses:**

1. As claimed in the abstract and conclusion, nighttime flare also degrades the performance of vision algorithms. Thus it may be better to include a simple experiment showing the benefits the dataset can bring to existing downstream vision algorithms, e.g., the flare-removed outputs result in better segmentation or detection performance.
2. It may be better to compare more existing flare removal methods rather than haze removal and image enhancement methods in the experiments part. It is good to include such methods for comparison but training more learning-based methods can better claim the effectiveness of the proposed dataset.
3. The writing and figures can be improved further, see the Clarify part below.

**Additional Feedback:**

No.

**Clarity:**

The overall writing of this paper is easy to follow but there are still many issues to address for a better paper.

1. According to the template and guidance of NIPS paper format, only the first letter of section title should be uppercase.
2. The fonts in figures are not unified. For example, the fonts used in Figure 1 and 2. are obviously different from other figures. Please keep the fonts of figures the same and follow the instructions of official template, including those in the suppl.
3. There should be blank space before the references in Line 59 an Line 60.
4. 'are' should be 'is' in Line 63.
5. 'of' may be missing in Line 71, after the word 'dataset'.
6. 'unique' should be 'uniquify in Line 117.
7. 'datase' should be 'dataset' in Line 223.
8. A period is missing in Line 249.
9. 'remove' should be 'removing' in Line 266.

Anyway, the paper should be carefully checked and improved carefully, only some issues are listed above.

**Documentation:**

It seems that the Datasheet for Dataset is not provided for this paper.

**Ethics:**

No.

**Relation To Prior Work:**

Yes.

**Summary And Contributions:**

This paper presents the first synthetic dataset for nighttime flare removal. Authors collected diverse nighttime flare images with different types of cameras and light sources and then render flares based on the observations and statistics. These rendered flares can be added to flare-free nighttime images to form paired nighttime flare-corrupted and flare-free data. Compared to the only daytime flare removal dataset, flares generated in this paper look more similar to real flares in night scenes and more diverse. Annotations of flare components are also provided in this dataset which is helpful for other interesting applications related to nighttime flare images. Experimental results claims the effectiveness of this synthetic nighttime flare removal dataset.

---

> ### Author Response · Authors · 2022-08-26
> **Response to Reviewer E5H2 (Part1)**
>
> We appreciate the constructive comments on our paper. The raised concerns are addressed as follows. We also updated our paper and supplementary material.
>
> > *It may be better to include a simple experiment showing the benefits the dataset can bring to existing downstream vision algorithms.*
>
> Thank you for the good advice, we are now conducting this experiment. We will update the results in the supplementary material soon.
>
> > *It may be better to compare more existing flare removal methods rather than haze removal and image enhancement methods in the experiments part. It is good to include such methods for comparison but training more learning-based methods can better claim the effectiveness of the proposed dataset.*
>
> Thank you for the valuable suggestions. Since flare removal is a new task [1], existing flare removal methods that aim to remove scattering and reflective flares are quite limited as stated in Section 2.  Although Qiao et al. [2] proposed a flare removal network, they did not release the official code and dataset. Thus, we only compare our method with the flare removal method of Wu et al. [1] which is most related to our task.
>
> The reasons why we choose to compare our methods with nighttime dehazing methods and nighttime image enhancement methods are stated in **Nighttime defogging and nighttime visibility enhancement** of Section 2.  In this new section, we have explained why these nighttime defogging methods can also alleviate the glare effect.  We also have added a brief introduction for these compared methods in line 296-300 of the updated paper.
>
> As suggested, we have added more recent image restoration baselines, including Restormer [6], MPRNet [5], HINet [3], and Uformer [4]  in the updated paper. For deep learning-based methods, we re-train these methods using the same training data proposed in our Flare7K dataset. The comparison results are presented in Table 2 of Section 5 of the updated paper. More visual results of these methods are shown in Figure 3 of the updated supplementary material. For your convenience, we list the quantitative results of different methods as follows.
>
> | Method        |      |       | Real  |       |      |       | Synthetic |       |
> | ------------- | ---- | ----- | ----- | ----- | ---- | ----- | --------- | ----- |
> |               |      | PSNR  | SSIM  | LPIPS |      | PSNR  | SSIM      | LPIPS |
> | Input         |      | 22.56 | 0.857 | 0.078 |      | 22.77 | 0.921     | 0.060 |
> | U-Net [7]     |      | 26.11 | 0.879 | 0.055 |      | 29.07 | 0.958     | 0.022 |
> | HINet [3]     |      | 26.74 | 0.882 | 0.048 |      | 29.97 | 0.959     | 0.021 |
> | MPRNet [5]    |      | 26.14 | 0.878 | 0.050 |      | 29.87 | 0.959     | 0.020 |
> | Restormer [6] |      | 26.28 | 0.883 | 0.054 |      | 29.45 | 0.950     | 0.025 |
> | Uformer [4]   |      | 26.98 | 0.890 | 0.047 |      | 30.47 | 0.965     | 0.017 |
>
> References:
>
> - [1] Yicheng Wu, Qiurui He, Tianfan Xue, Rahul Garg, Jiawen Chen, Ashok Veeraraghavan, and Jonathan T. Barron, “How to train neural networks for flare removal.” in *IEEE International Conference on Computer Vision*, 2021.
>
> - [2] Xiaotian Qiao, Gerhard P. Hancke, and Rynson W. H. La,  “Light source guided single-image flare removal from unpaired data.” in *IEEE International Conference on Computer Vision*, 2021.
>
> - [3] Liangyu Chen,  Xin Lu, Jie Zhang, Xiaojie Chu, and Chengpeng Chen, "HINet: Half instance normalization network for image restoration." in *IEEE Conference on Computer Vision and Pattern Recognition Workshops*, 2021.
>
> - [4] Zhendong Wang, Xiaodong Cun, Jianmin Bao, Wengang Zhou, Jianzhuang Liu, and Houqiang Li, “Uformer: A general u-shaped transformer for image restoration.” in *IEEE Conference on Computer Vision and Pattern Recognition*, 2022.
>
> - [5] Syed Waqas Zamir, Aditya Arora, Salman Khan, Munawar Hayat, Fahad Shahbaz Khan, Ming-Hsuan Yang, and Ling Shao,  “Multi-stage progressive image restoration.” in *IEEE Conference on Computer Vision and Pattern Recognition*, 2021.
>
> - [6] Syed Waqas Zamir,  Aditya Arora, Salman Khan, Munawar Hayat, Fahad Shahbaz Khan, and Ming-Hsuan Yang, "Restormer: Efficient transformer for high-resolution image restoration." in *IEEE Conference on Computer Vision and Pattern Recognition*, 2022.
>
> - [7] Olaf Ronneberger, Philipp Fischer, and Thomas Brox, “U-Net: Convolutional networks for biomedical image segmentation.” in *International Conference on Medical Image Computing and Computer-Assisted Intervention*, 2015.
>
>
>
> > *The writing and figures can be improved further, see the Clarify part below.*
>
> Thank you for the suggestions. We have rewritten the paper and supplementary material and redrawn some figures in the updated paper. We also have fixed the typos. The revised parts are marked in blue.
>
> > *It seems that the Datasheet for Dataset is not provided for this paper.*
>
> We have added a datasheet for our dataset. Please check it in Section 7 of the updated supplementary material.

---

> > ### Author Response · Authors · 2022-08-29
> > **Response to Reviewer E5H2 (Part2)**
> >
> > > *As claimed in the abstract and conclusion, nighttime flare also degrades the performance of vision algorithms. Thus it may be better to include a simple experiment showing the benefits the dataset can bring to existing downstream vision algorithms, e.g., the flare-removed outputs result in better segmentation or detection performance.*
> >
> > Thank you so much for the advice. We have stated that *"Taking nighttime driving with stereo cameras as an example, the scattering flare may be misestimated as close obstacles by stereo matching algorithms."* in lines 26-27 of the main paper. To suggest that our dataset can bring benefits to existing downstream vision algorithms, we have added an experiment on the flare removal algorithm's influence on stereo matching results. In this experiment, we use ZED stereo camera to capture real-world paired flare-corrupted images. Then we test whether the flare removal algorithm trained on our dataset can alleviate the mismatching caused by lens flares. Our result shows that the flare removal model trained on our dataset can boost the robustness of the stereo matching algorithm significantly. The experiment details are shown in Section 3 of the updated supplementary material. The visual comparison is shown in Figure 5 of the updated supplementary material.
> >
> > Due to the limited time, we only test one downstream vision algorithm. In the camera-ready version, we will add more experiments to show the benefits of our flare removal dataset.

---

### Official Review · Reviewer_eGou · 2022-07-27
**A Nighttime Flare Removal Dataset.**

**Rating:** 5
**Confidence:** 5

**Strengths:**

This proposed dataset is mainly designed to improve the quality of nighttime images and might help the community to strengthen the research.

**Weaknesses:**

1. Some no-reference metrics should be used in the Table 2 of the manuscript.
2. In th table 2, the benchmarking study is not convcing due to lack some the general methods for image restoration such as MPRNet, etc.


**Additional Feedback:**

Stated in Weaknesses.

**Clarity:**

The paper is well written and well organized.


**Correctness:**

Yes, this dataset is constructed in a sound way and I think this paper offers a good building process for future restoration benchmarks.

**Documentation:**

Yes

**Ethics:**

There is no ethical concern in this paper.

**Relation To Prior Work:**

It is introduced well in section 2 of the manuscript.

**Summary And Contributions:**

This work aims at advancing nighttime flare removal and proposes a new dataset called Flare7K. Based on the proposed dataset, extensive experiments are conducted, which demonstrates that the proposed dataset can complement the diversity of existing flare datasets and push the frontier of nighttime flare removal.

---

> ### Author Response · Authors · 2022-08-26
> **Response to Reviewer eGou (Part1)**
>
> We sincerely thank you for reviewing our paper and providing us valuable feedback. We have addressed your concerns as below. The revised parts have been marked in blue in the updated paper and supplementary material.
>
> > *Some no-reference metrics should be used in Table 2 of the manuscript.*
>
> Thank you for your suggestions. Actually, we tried to use both full-reference metrics and non-reference metrics to evaluate the performance of different methods. For the non-reference metric, we used PI (Perceptual Index) [1], MUSIQ [2], NIQE [3], and NIMA [4]  on the real-world test data. However, we found that existing non-reference metrics cannot accurately reflect the quality of different results.  We also list the results of non-reference metrics as follows.
>
> |               | PSNR$\uparrow$ | SSIM$\uparrow$ | LPIPS$\downarrow$ | MUSIQ$\uparrow$ | NIQE$\downarrow$ | NIMA$\uparrow$ | PI $\downarrow$ |
> | ------------- | :------------: | :------------: | :---------------: | :-------------: | :--------------: | :------------: | --------------- |
> | Input         |     22.561     |     0.8567     |      0.0776       |      59.34      |      4.563       |   **4.793**    | 3.606           |
> | Ground truth  |       /        |       /        |         /         |    **59.95**    |      4.505       |     4.689      | 3.494           |
> | Wu et al. [5] |     24.613     |     0.8713     |      0.0598       |      59.51      |    **4.339**     |     4.675      | **3.423**       |
> | Ours          |   **26.107**   |   **0.8785**   |    **0.0549**     |      59.31      |      4.505       |     4.656      | 3.479           |
>
> Although some non-reference metrics are effective for natural images such as images with noise or JPEG artifacts, they fail in nighttime flare data. As shown, NIQE, NIMA, and PI cannot effectively represent the quality of ground truth images. Moreover, when observing the MUSIQ score of each image, we found the MUSIQ scores of some input images are higher than those of the corresponding ground truth images. Thus, MUSIQ cannot also be used as the non-reference metric in our task.  Thus, we only use full-reference metrics in our experiments as the ground truth images are available. We thank the good suggestions again.
>
>
>
> Reference:
>
> - [1] Yochai Blau, Roey Mechrez, Radu Timofte, Tomer Michaeli, and Lihi Zelnik-Manor, "The 2018 PIRM challenge on perceptual image super-resolution." in *European Conference on Computer Vision Workshops*, 2018.
> - [2] Junjie Ke, Qifei Wang, Yilin Wang, Peyman Milanfar, and Feng Yang, "MUSIQ: Multi-scale image quality transformer." in *IEEE International Conference on Computer Vision*, 2021.
> - [3] Anish Mittal, Rajiv Soundararajan, and Alan C. Bovik, “Making a ‘completely blind’ image quality analyzer.” *IEEE Signal Processing Letters*, 2013.
> - [4] Talebi, Hossein, and Peyman Milanfar, “NIMA: Neural image assessment.” *IEEE Transactions on Image Processing*, 2018.
> - [5] Yicheng Wu, Qiurui He, Tianfan Xue, Rahul Garg, Jiawen Chen, Ashok Veeraraghavan, and Jonathan T. Barron, “How to train neural networks for flare removal.” in *IEEE International Conference on Computer Vision*, 2021.

---

> > ### Author Response · Authors · 2022-08-26
> > **Response to Reviewer eGou (Part2)**
> >
> > > *In table 2, the benchmarking study is not convincing due to the lack some the general methods for image restoration such as MPRNet, etc.*
> >
> > Thank you for your constructive comments. As suggested, we have added more recent deep learning-based image restoration methods including HINet [6], Uformer [7], MPRNet [8], and Restormer [9] for comparison. We have detailed the **Experiment Setting** in Section 5 and added the results in Table 2 of the updated paper. For your convenience, we present the quantitative results as follows. More details and visual results can be found in Section 2 of the updated supplementary material.
> >
> > | Method        |      |       | Real  |       |      |       | Synthetic |       |
> > | ------------- | ---- | ----- | ----- | ----- | ---- | ----- | --------- | ----- |
> > |               |      | PSNR  | SSIM  | LPIPS |      | PSNR  | SSIM | LPIPS |
> > | Input         |      | 22.56 | 0.857 | 0.078 |      | 22.77 | 0.921 | 0.060 |
> > | U-Net [10]    |      | 26.11 | 0.879 | 0.055 |      | 29.07 | 0.958 | 0.022 |
> > | HINet [6]     |      | 26.74 | 0.882 | 0.048 |      | 29.97 | 0.959 | 0.021 |
> > | MPRNet [8]    |      | 26.14 | 0.878 | 0.050 |      | 29.87 | 0.959 | 0.020 |
> > | Restormer [9] |      | 26.28 | 0.883 | 0.054 |      | 29.45 | 0.950 | 0.025 |
> > | Uformer [7]   |      | 26.98 | 0.890 | 0.047 |      | 30.47 | 0.965 | 0.017 |
> >
> > Reference:
> >
> > - [6] Liangyu Chen,  Xin Lu, Jie Zhang, Xiaojie Chu, and Chengpeng Chen, "HINet: Half instance normalization network for image restoration." in *IEEE Conference on Computer Vision and Pattern Recognition Workshops*, 2021.
> > - [7] Zhendong Wang, Xiaodong Cun, Jianmin Bao, Wengang Zhou, Jianzhuang Liu, and Houqiang Li, “Uformer: A general u-shaped transformer for image restoration.” in *IEEE Conference on Computer Vision and Pattern Recognition*, 2022.
> > - [8] Syed Waqas Zamir, Aditya Arora, Salman Khan, Munawar Hayat, Fahad Shahbaz Khan, Ming-Hsuan Yang, and Ling Shao,  “Multi-stage progressive image restoration.” in *IEEE Conference on Computer Vision and Pattern Recognition*, 2021.
> > - [9] Syed Waqas Zamir,  Aditya Arora, Salman Khan, Munawar Hayat, Fahad Shahbaz Khan, and Ming-Hsuan Yang, "Restormer: Efficient transformer for high-resolution image restoration." in *IEEE Conference on Computer Vision and Pattern Recognition*, 2022.
> > - [10] Olaf Ronneberger, Philipp Fischer, and Thomas Brox, “U-Net: Convolutional networks for biomedical image segmentation.” in *International Conference on Medical Image Computing and Computer-Assisted Intervention*, 2015.

---

### Author Response · Authors · 2022-08-29
**Kind Reminder**

We appreciate all the reviewers for their valuable time and constructive comments. It enables us to substantially improve the quality of our paper and proposed dataset.

In our revised paper, all the issues that the reviewers raised have been addressed. The changes we made can be summarized as follows:

- To improve the readability of our paper, we have rewritten the Section 4 and Section 5 of the main paper. Also, we have added citations for necessary references and a new paragraph about nighttime visual enhancement and nighttime dehazing in the related work.
- We build up a benchmark for state-of-the-art deep learning-based image restoration methods on our dataset.
- We have set up a Github repository: https://github.com/ykdai/Flare7K to release our dataloader function and flare-corrupted image generation script. When our paper is accepted, we will release the training code and pre-trained models.
- We have increased the number of synthetic/real test data from 20 to 100 for each.
- To show that our flare removal method can also improve the performance of downstream tasks, we have added an experiment to explore the effects of flare removal model trained on our dataset for stereo matching.
- We have added a datasheet in the supplementary material to show more details of our dataset.

Since it is close to the end of the discussion period, we would like to kindly remind the reviewers to confirm whether our changes solved your concerns. If you have any further questions, do not hesitate to reply to our responses. Without your help, we could not build this useful, reasonable, and robust dataset. Thank you again for all the efforts that you have made.

Best wishes,

Authors of paper Flare7K: A Phenomenological Nighttime Flare Removal Dataset

---

### Meta-Review · Area_Chair_8rhT · 2022-09-07

**Recommendation:** Accept
**Confidence:** 5

**Metareview:**

This paper propose a new nighttime flare hazard dataset. The synthetic dataset enables to obtain ground truth which is otherwise very difficult in real scenes. All reviewers are satisfied with the quality of the paper. Only one reviewer recommends marginal below acceptance, and the concerns are minor. The concerns has been carefully addressed by the authors through the discussion. All other concerns have also been carefully addressed and the paper has been rewritten accordingly. Therefore this paper is suitable for acceptance.

---

### Decision · Program_Chairs · 2022-09-16

Accept